# The sperm hook as a functional adaptation for migration and self-organized behavior

**Heungjin Ryu[1,2]\*[†], Kibum Nam[1†], Byeong Eun Lee[3†], Yundon Jeong[1†], Seunghun Lee[1], Jeongmo Kim[1], Young-Min Hyun[4], Jae-Ick Kim[3]\*, Jung-Hoon Park[1]\***

[1]Department of Biomedical Engineering, Ulsan National Institute of Science and Technology, Ulsan, Republic of Korea; [2]Department of Social Informatics, Kyoto University, Kyoto, Japan; [3]Department of Biological Sciences, Ulsan National Institute of Science and Technology, Ulsan, Republic of Korea; [4]Department of Anatomy, Yonsei University College of Medicine, Seoul, Republic of Korea

**\*For correspondence:**
ryu.heungjin.26v@kyoto-u.jp (HR);
jikim220@unist.ac.kr (JIK);
jh.park@unist.ac.kr (JHP)

[†]These authors contributed equally to this work

**Competing interest:** The authors declare that no competing interests exist.

## eLife Assessment

This study uses ex vivo live imaging of uteri post-mating to test the role of the sperm hook in the house mouse sperm in sperm movement that would be interesting to evolutionary biologists. The significance of the work is **useful** as live imaging can reveal information not seen in fixed images. The strength of evidence is **incomplete** as they cannot directly test the role of the sperm hook in facilitating movement along the uterine wall.

**Abstract** In most murine species, spermatozoa exhibit a falciform apical hook at the head end. The function of the sperm hook is not yet clearly understood. In this study, we investigate the role of the sperm hook in the migration of spermatozoa through the female reproductive tract in *Mus musculus* (C57BL/6), using a deep tissue imaging custom-built two-photon microscope. Through live reproductive tract imaging, we found evidence indicating that the sperm hook aids in the attachment of spermatozoa to the epithelium and facilitates interactions between spermatozoa and the epithelium during migration in the uterus and oviduct. We also observed synchronized sperm beating, which resulted from the spontaneous unidirectional rearrangement of spermatozoa in the uterus. Based on live imaging of spermatozoa-epithelium interaction dynamics, we propose that the sperm hook plays a crucial role in successful migration through the female reproductive tract by providing anchor-like mechanical support and facilitating interactions between spermatozoa and the female reproductive tract in the house mouse.

## Introduction

Since the pioneering work by *Parker, 1970*, postcopulatory sexual selection has been recognized as an important driving force that shapes male sperm and female reproductive track characteristics. Post-copulatory sperm competition over gamete and cryptic female choice are equivalent to male-male competition and female choice in the pre-copulatory sexual selection. Sperm competition occurs when spermatozoa from more than one male coincide within the female reproductive tract in a time window where they have a chance to fertilize the same egg (*Parker, 1970*). The evolutionary consequences of the sperm competition for the male traits, including baculum complexity, sperm shape and behavior, are demonstrated in a wide range of animals from insects to mammals (*Anderson et al.,*

*2005*; *Lange et al., 2013*; *André et al., 2020*). Such male adaptations secure sperm delivery to the female reproductive tract and influence the sequential mating of the already-mated female with other males. Cryptic female choice is a more complicated process to define as it is achieved at the various stages of the post-mating fertilization process that also occurs inside the female reproductive tract. However, post-copulatory cryptic female choice is evident where a female can exert an influence on ejaculates or even fertilized eggs that can influence the male's reproductive success (*Eberhard, 1996*; *Firman et al., 2017*; *Roberts et al., 2012*).

Postcopulatory sexual selection results in the co-evolution of the male and female traits that impact the reproductive strategies of the opposite sex (*Eberhard and Lehmann, 2019*; *Greff and Parker, 2000*). For example, copulatory plugs formed by male semen hinder the sequential mating of the female with a second male and diminish the reproductive outcome of the second male (*Mangels et al., 2016*; *Sutter and Lindholm, 2016*). Females, on the other hand, by removing the copulatory plug, can counteract the male strategy (*Koprowski, 1992*). When fluid in female reproductive tract is spermicidal, male seminal fluid neutralises the spermicidal medium and secures sperm survival inside the female reproductive tract (*Holman and Snook, 2008*; *Holman and Snook, 2006*).

Among mammals, rodent species exhibit various sperm morphological and behavioral characteristics. A notable morphological feature of murine sperm is the apical hook resulting in an asymmetrical falciform head shape that is found in most of the murine rodents (*Breed, 2004*; *Roldan et al., 1992*). The functional significance of the sperm hook that causes head asymmetry is still under debate in the field of mouse reproductive ecology. Currently, two main hypotheses attempt to explain the function of the sperm hook. One is that the sperm hook plays a crucial role in sperm competition by aiding sperm linkage (sperm train formation) that enhances straighter and faster sperm forward progression – the sperm cooperation hypothesis (*Fisher and Hoekstra, 2010*; *Moore et al., 2002*). The other hypothesis is that the sperm hook facilitates sperm-epithelium interactions in the female reproductive tract, playing a significant role in sperm migration – the migration hypothesis (*Smith and Yanagimachi, 1990*; *Firman and Simmons, 2009*). The sperm cooperation hypothesis is supported by the pioneering discovery of sperm linkage known as sperm trains (*Moore et al., 2002*) and suggests that sperm trains facilitate faster or straighter sperm swimming (*Fisher and Hoekstra, 2010*; *Moore et al., 2002*). However, other studies could not find supporting evidence of sperm cooperation by the sperm train or morphological changes in sperm hooks concerning the degree of sperm competition (*Firman and Simmons, 2009*; *Hook et al., 2021*). These researchers rather suggest that the sperm hook plays a crucial role in sperm migration by interacting with epithelia in the female reproductive tract.

Since thick muscle layers comprise the outer part of the rodent female reproductive tract, direct observation of spermatozoa inside the female reproductive tract is a challenging feat. Therefore, previous studies focused on the oviduct where muscle layers are thin and transparent, using bright-field, fluorescence or confocal microscopy (*Ishikawa et al., 2016*; *Qu et al., 2021*; *Suarez, 1987*). In this study, we developed an ex-vivo observation system based on a custom-built two-photon microscope (*Figure 1—figure supplement 1*). This system enables the observation of spermatozoa and their behavior inside the live female reproductive tract. While two-photon microscopy is currently the method of choice for live imaging of deep tissues (*Helmchen and Denk, 2005*; *Benninger and Piston, 2013*; *Keller, 2013*), it has seldom been used for studying animal reproductive ecology. Here, we demonstrate high-resolution deep-tissue imaging that allows us to observe and track sperm movement inside the female reproductive tract, including the uterus, to realise real-time tracking of spermatozoa and their migration.

Using this technology, we report newly discovered sperm behavior and aggregation patterns that suggest various roles of the sperm hook in sperm migration inside the female reproductive tract. Based on live imaging of sperm dynamics inside the live female reproductive tract, we provide new observations and insights that suggest functions of the sperm hook that help sperm migration and cooperation. With our new observation system and results, we hope to contribute to the field of postcopulatory sexual selection in rodents by advancing methodological progress and stimulating discussion and future research on the function of sperm hook in murine rodents.

## Results

To observe mouse sperm behavior in the female reproductive tract, we mated wild-type *Mus musculus* (C57BL/6) females with transgenic male mice (*Supplementary file 1a, b*) that express DsRed at the

sperm mid-piece (mitochondria) and eGFP in the sperm acrosome (*Hasuwa et al., 2010*). The female mice were euthanized between 0.5- and 3 hr post-mating, and their reproductive tract along with the copulatory plug was excised and transferred to a sterilized Petri dish. We then conducted ex-vivo imaging of the live reproductive tract using our two-photon microscope (*Figure 1—figure supplement 1*). *Figure 1A* is a diagram of the female reproductive tract that labels the name of each part of the tract. All observations were typically completed within 3 hr and did not exceed 6 hr post-euthanasia. Despite this observation period being longer than recommended for conventional rat transplantation surgery (*Díaz-García et al., 2013*), we noted live uterine movements throughout the entire observation period. When samples were transferred to incubators preheated to 37°C with 5% $CO_2$ following observation, uterine movement persisted even 24 hr post-excision. We used Fiji (*Schindelin et al., 2012*) and R 4.3.3 (*R Development Core Team, 2024*) for image processing and statistical analysis.

## The role of sperm hook in migration inside the uterus

We first observed sperm movement in the uterus, where most ejaculated spermatozoa are located and initial sperm selection occurs. In the uterus, we observed vigorous fluid flow caused by uterine contraction and relaxation. Consequently, most spermatozoa in the uterus were carried by this flow, adhering to the inherent flow dynamics within the uterus (*Figure 1—video 1A*). However, when the flow temporarily ceased, spermatozoa located near the uterine wall exhibited greater activity (*Figure 1—video 1B, C*). When a spermatozoon encounters the uterine wall (epithelium) during migration, the sperm hook interacted with the wall and acts as a pivot (*Yu et al., 2023*) that influences the direction of sperm travel (*Figure 1B and C*). *Figure 1—video 2A* shows two distinct types of sperm movement when transitioning from the uterus volume to the uterine wall: pro-wall-hook and anti-wall-hook directional movement (*Figure 1C*). Upon reaching the uterine wall, instead of randomly deflecting in various directions, sperm preferentially altered their heading direction such that their apical hook would face the uterine wall (pro-wall-hook direction).

To test whether the hook influences the direction of migration, we tracked spermatozoa that travelled from the uterus volume to the wall in sequentially acquired images. We found that 52 out of 63 spermatozoa (82.54%) changed their migration direction towards the pro-wall-hook direction after reaching the uterine wall (*Figure 1D*). The remaining 11 spermatozoa followed the anti-wall-hook direction. A binomial test confirmed that this tendency was statistically significant (one-tailed, p<0.001, 95% CI: 0.73, 1.00). Furthermore, when spermatozoa migrated along the uterine wall, they exhibited a tapping-like behavior in which spermatozoa tapped their hook against the epithelium when oriented in a pro-wall-hook direction (*Figure 1—video 2B*). In contrast, when the sperm hook faces the luminal space, spermatozoa could not migrate along the wall and their migration trajectories followed the anti-wall-hook direction, resulting in the movement away from the wall (*Figure 1D*; *Figure 1—video 2A*). In addition to the advantage of straighter trajectories, fluid flows are slower at the wall than at the centre of the lumen in the uterus (*Zaferani et al., 2019*). Therefore, migration along the uterine wall will help spermatozoa to reach the entrance of the intramural utero-tubal junction (UTJ), also called colliculus tubarius (CT), where one of the most important sperm selection processes takes place (*Nakanishi et al., 2004*; *Qu et al., 2021*).

## Sperm anchoring and migration kinetics in the uterus

We observed that the sperm head was attaching to the epithelium of the CT and uterus (*Figure 1—video 3A, B*) when spermatozoa reached the CT. The sperm hook was affixed to the epithelium, thereby securing the spermatozoa on the epithelia. We defined this securing of the spermatozoa on the epithelia by the sperm hook as an anchor-like function of the hook (*Figure 1—video 3A, B*). This anchoring also helped prevent spermatozoa from being swept away by mucosal flow (*Figure 1E*; *Figure 1—video 3B*). When spermatozoa are attached to the CT and squeezed through other spermatozoa by hooking, the apical hook always faces the epithelium (*Figure 1F*; *Figure 1—video 3A*). Once spermatozoa successfully clung onto the CT, anchoring to the epithelia prevented them from being pushed out by competing spermatozoa or from being swept away by fluid flow (*Figure 1—video 3B*).

To compare sperm migration kinetics during sperm swimming relative to their distance from the uterine wall, we tracked sperm migration trajectories by employing the TrackMate plugin in ImageJ

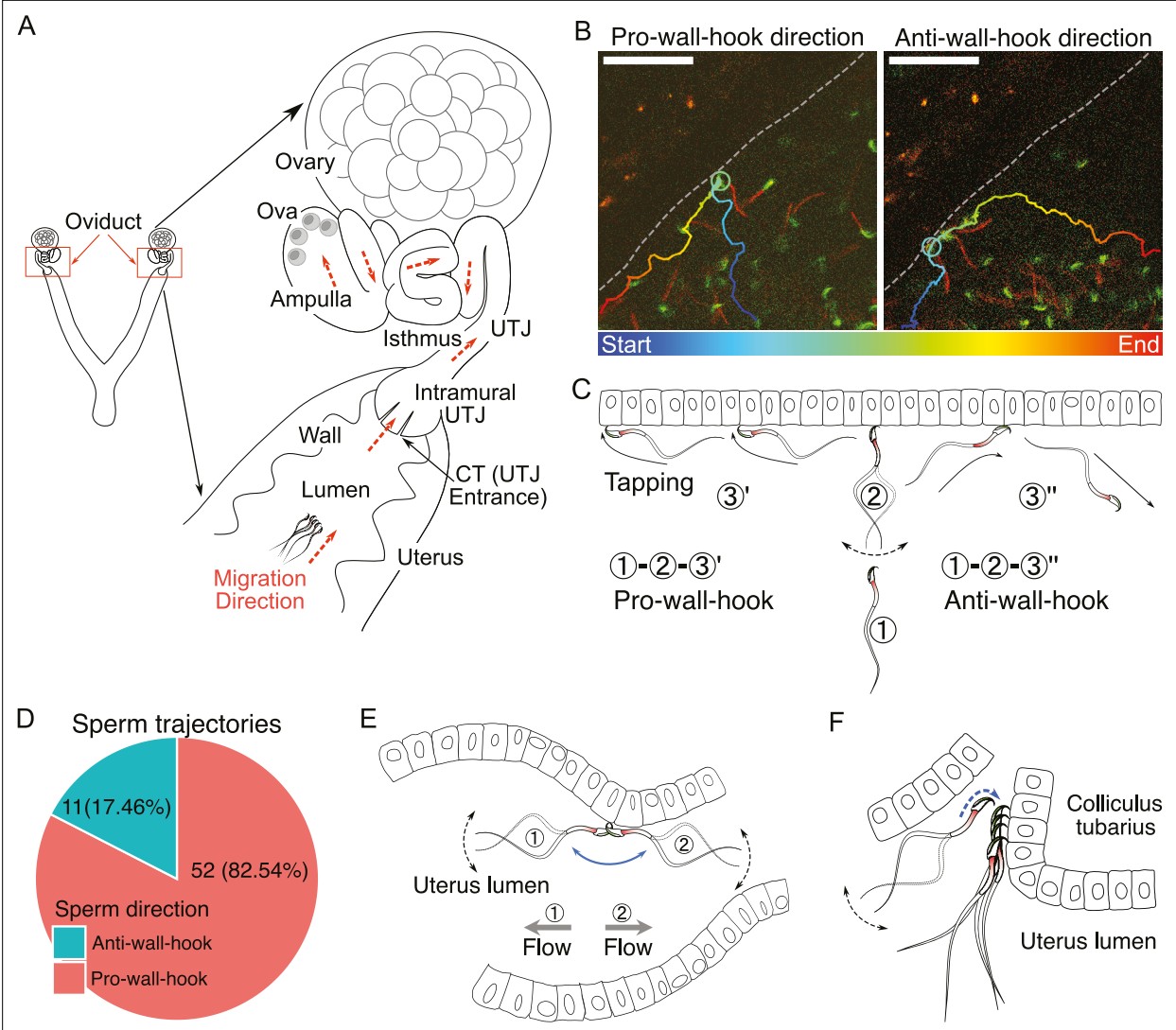

**Figure 1.** The structure of the female reproductive tract and various sperm behaviors and interactions with epithelia in the uterus. (**A**) An illustration of the female reproductive tract and sperm migration direction (arrows in orange colour) inside the tract. Ova (egg cells) are in the ampulla. (**B**) Spermatozoa alter their travel direction based on their head orientation upon reaching the uterine wall (sperm hook functions as a pivot, *Figure 1—video 2A*). Left image shows a sperm trajectory exhibiting a pro-wall-hook direction, and the right image shows an anti-wall-hook direction. The trajectory is depicted in colors representing different time points. Dashed lines are guides to the eye to visualize the uterus wall. Scale bar: 50 µm. (**C**) Sperm moving direction changes over time when they reach the uterine epithelia. Numbers indicate sequences of sperm movement. (1) During sperm migration, they reach the uterine wall. (2) After reaching the uterine wall, they beat several times while facing their head towards the uterine wall. (3) Sperm beating results in a change in sperm orientation (1–3': pro-wall-hook, 1–3": anti-wall-hook). The pro-wall-hook orientation coincides with sperm travelling along the wall and the anti-wall-hook orientation coincides with sperm moving away from the wall. (**D**) When spermatozoa reach the uterine wall, their trajectories predominantly follow the pro-wall-hook direction, where the sperm hook is directed towards the uterine wall. (**E**) The sperm hook assists a spermatozoon in anchoring to the epithelia (hook as an anchor). This anchoring facilitates sperm attachment to the uterine and utero-tubal junction (UTJ) epithelium and prevents spermatozoa from being swept away by internal flow caused by peristaltic movement. (**F**) The sperm hook and thin sperm head aid spermatozoa in squeezing through the sperm-crowded UTJ entrance (colliculus tubarius, CT) and attaching to the epithelium by acting as an anchor (*Figure 1—video 3*). Note that in our experiments, we do not observe the principal and terminal pieces of spermatozoa due to a lack of fluorescence in our animal model. The observed acrosome and midpiece are shown coloured in the schematic figures.

The online version of this article includes the following video and figure supplement(s) for figure 1:

**Figure supplement 1.** Schematic diagram of the custom-built 2PSLM.

**Figure 1—video 1.** Sperm migration in the uterus.

https://elifesciences.org/articles/96582/figures#fig1video1

**Figure 1—video 2.** The sperm hook helps sperm to determine migration directions in the uterus.

*Figure 1 continued on next page*

*Figure 1 continued*

https://elifesciences.org/articles/96582/figures#fig1video2

**Figure 1—video 3.** Sperm use their hook as an anchor to be attached to the uterine epithelium.

https://elifesciences.org/articles/96582/figures#fig1video3

(*Ershov et al., 2022*; *Tinevez et al., 2017*). In total, we were able to track 694 sperm trajectories. After successful tracking using the customized tracking option in TrackMate, we computed various sperm kinetic parameters. These parameters included the curvilinear velocity (VCL), straight-line velocity (VSL), and linearity of forward progression (LIN), which are commonly used in computer-assisted sperm analysis, CASA (*Amann and Waberski, 2014*). Briefly, VCL was calculated as the total distance travelled divided by total travel time, VSL as the distance between initial and final positions of the sperm trajectory divided by total travel time, and LIN as the ratio of VSL to VCL, which can range from 0 to 100%, with 100% representing a perfectly straight line. We also introduced a new kinetic parameter called straight line-to-sideward movement ratio (SWR), defined as track displacement of a sperm trajectory divided by maximum sideward movement distance. Refer to *Figure 2—figure supplement 1* for our definition of the uterus wall and a schematic description of the sperm migration kinetic parameters, as well as Methods for more details.

Our sperm tracking analysis revealed that spermatozoa located close to the uterine wall moved faster, exhibiting higher VCL and VSL (*Figure 2A*; *Supplementary file 1c*; *Figure 1—video 1B, C*).

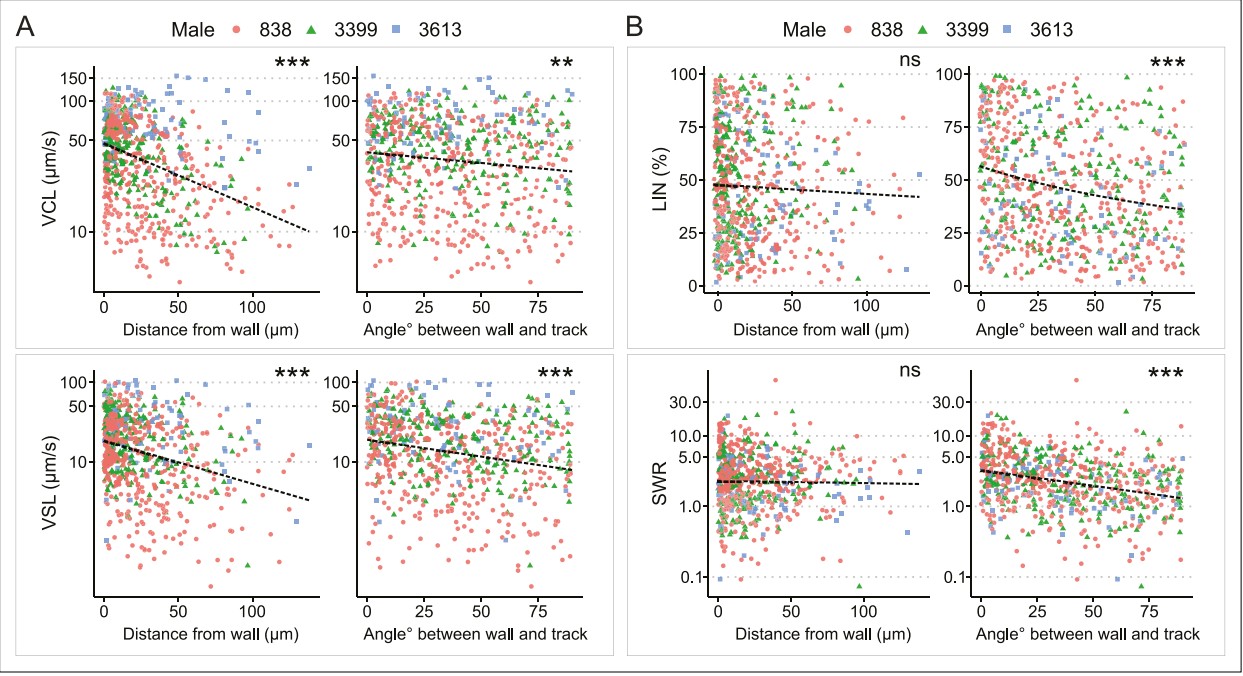

**Figure 2.** Analysis of sperm kinetic parameters relative to the distance and angle between the sperm trajectory and uterine wall. (**A**) Both curvilinear velocity (VCL) (top) and straight-line velocity (VSL) (bottom) decreased with an increase in the distance between the sperm trajectory and the uterine wall. Similarly, VCL and VSL decreased as the angle between the sperm trajectory and uterine wall increased. (**B**) The distance between the sperm trajectory and uterine wall did not significantly affect linearity of forward progression (LIN) (top left) and SWR (bottom left). However, both LIN and straight line-to-sideward movement ratio (SWR) decreased when the angle between the sperm trajectory and uterine wall increased. The total number of sperm trajectories is 694. Data from different males are represented in different colors and shapes. The dotted lines indicate regression lines from simple regressions to aid visual interpretation. Check model estimates for more details and precise interpretation of the models (*Supplementary file 1*). The y-axis of each figure is displayed in the log-scale except for LIN. Images for sperm tracking were acquired around the area labeled as 'Wall' in the uterus in *Figure 1A*. Sperm trajectories used in this analysis are presented in *Figure 1—video 1B*. The effect of the two random variables (males and females) are visualized in *Figure 2—figure supplement 2*. ***: $p < 0.001$, **: $p < 0.01$, *: $p < 0.05$, ns: not significant.

The online version of this article includes the following figure supplement(s) for figure 2:

**Figure supplement 1.** Uterus wall and parameters that were used to measure sperm migration speed and linearity.

**Figure supplement 2.** Visualization of male and female effects (two random effects) on sperm migration kinematic parameters.

However, LIN and SWR did not significantly vary depending on the sperm's distance from the uterus wall (*Figure 2B*). In contrast to the non-significant changes in LIN and SWR relative to the distance from the uterine wall, when spermatozoa swam parallel to the wall, they not only moved faster (higher VCL and VSL; *Figure 2A*), but also followed a straighter path (higher LIN and SWR; *Figure 2B*). Given that the internal fluid flow around the uterine wall is slower (*Zaferani et al., 2019*), and considering the hydrodynamic factors that attract spermatozoa to the wall when swimming near it (*Alvarez et al., 2014*; *Elgeti et al., 2010*), migration along the wall would be an efficient strategy for sperm movement within the uterus. We also examined the impact of both individual males (coded as 'Male ID') and females (coded as 'Date') on the sperm kinetic parameters in all models by visualizing the effects of these two random variables (*Figure 2—figure supplement 2*). We confirmed that the effects of the two random variables were consistent across all significant models, indicating the validity of our model estimations.

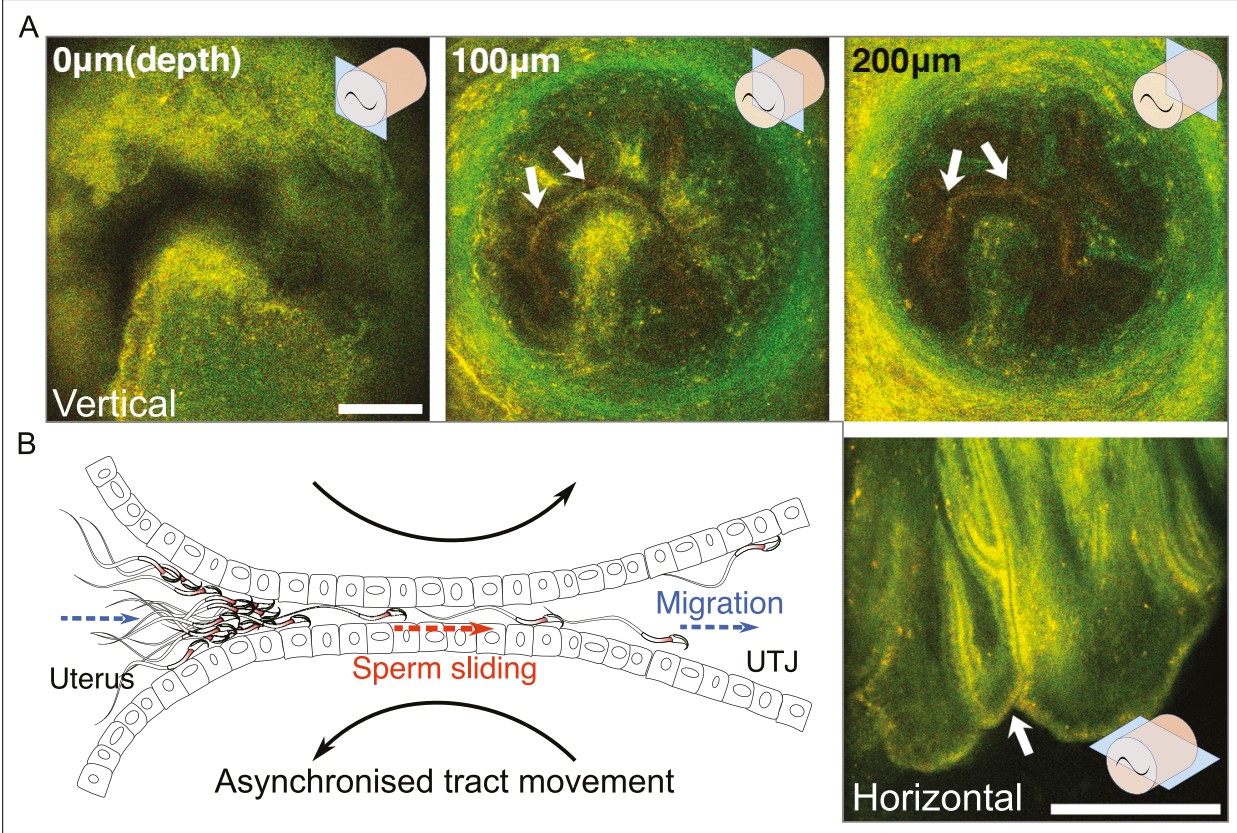

**Figure 3.** The structure of the entrance of intramural utero-tubal junction (UTJ) and sperm passage. (**A**) The upper three images are the vertical view of the colliculus tubarius (CT) (entrance to the UTJ) of the intramural UTJ from an unmated female. The bottom right image is a horizontal view of the intramural UTJ. There are only a few small gaps, indicated by arrows, between mucosal folds, which control sperm migration into the UTJ from the uterus (*Figure 3—video 1*). Illustrations at upper and lower right conners indicate imaging planes (blue rectangles) and the intramural UTJ (orange cylindrical structure) with the CT (black tilde). Scale bar: 100 µm, Arrow line thickness: 10 µm for top three images, 5 µm right bottom image. (**B**) Asynchronized movement of mucosal folds (sliding against each other) at the CT due to uterine and UTJ contractions enables spermatozoa to slide into the intramural UTJ from the uterus (*Figure 3—video 2*). Two dashed arrows (in blue) indicate the direction of sperm migration from the uterus to the UTJ, and the dashed arrow in the centre (in orange) indicates the direction of sperm sliding in the intramural UTJ. The two curved black arrows indicate a asynchronized (opposite) movement of confronting mucosal folds in the intramural UTJ.

The online version of this article includes the following video and figure supplement(s) for figure 3:

**Figure supplement 1.** A hypothetical model for sperm migration from the uterus to utero-tubal junction (UTJ).

**Figure 3—video 1.** The entrance of intra-mural utero-tubal junction (UTJ) (or colliculus tubarius, CT) in the uterus has small spacing (almost closed inter-fold gaps) for mouse sperm to pass through.

https://elifesciences.org/articles/96582/figures#fig3video1

**Figure 3—video 2.** Sperm behaviors and the movement of mucosal folds at the entrance of utero-tubal junction (UTJ) (colliculus tubarius, CT).

https://elifesciences.org/articles/96582/figures#fig3video2

## Structure of UTJ and sperm behavior at the entrance of intramural UTJ (CT)

Upon sacrificing and optical clearing the tissue, we confirmed that the entrance to the intramural UTJ (CT) in the uterus consists of nearly closed narrow gaps between mucosal folds (*Figure 3A*). These narrow gaps extended to about 100 μm deeper from the entrance (*Figure 3—video 1A, B*), and only a few spermatozoa could pass through a gap at a time (*Figure 3A*; *Figure 3—video 1C*). We were not able to find evidence of passive sperm carriage, such as upsuck-like sperm carriage (*Baker and Bellis, 1993*), caused by peristaltic movement from the uterus into the UTJ in real-time live images (*Figure 3—video 2A*). Therefore, we concluded that fluid flow induced by uterine and oviduct contraction is not a major driving force for sperm entry to the UTJ through the CT.

This conclusion raises a question about how spermatozoa enter the UTJ through the CT if the UTJ entrance is nearly closed. We found head-directional sliding of spermatozoa in the almost closed inter-luminal spaces between the uterus and intramural UTJ (*Figure 3—video 2B*). We did not observe any sliding of spermatozoa that were directs to the tail direction. This one-directional head-forward sliding in a very narrow inter-luminal space suggests the role of the sperm hook as an anchor that prevents backwards slipping by the squeezing movement from the contraction of the uterus. Such one-directional sliding is also a plausible way of sperm migration from the uterus to UTJ through the narrow gap at the CT. We also observed that when muscle contraction and relaxation occurred at the uterus and oviduct, the surfaces of two confronting mucosal folds in intramural UTJ slid against each other in opposite directions (*Figure 3B*; *Figure 3—video 2B*). Such an opposite directional movement of mucosal folds, for example, will occur when the uterus bends to the left due to muscle contraction on its left side and relaxation of its right side (*Figure 3—figure supplement 1*). As shown in *Figure 3—video 2*, the opposite movement of mucosal folds makes space between mucosal folds at the CT, providing an opportunity for nearby attached spermatozoa to enter the intramural UTJ (*Figure 3—figure supplement 1*). Although the application of further experimental approaches will be necessary, occasional fluctuations in the size of the luminal space between mucosal folds, caused by peristaltic movements, together with the head-directional sliding of the spermatozoa, may provide an opportunity for spermatozoa to pass through the CT.

We also observed unidirectional sperm clustering as a result of spontaneous sperm re-arrangement during sperm beating along the uterine wall (*Figure 4A* and *Figure 4—video 1A, B*). Such unidirectional sperm clustering and their successive beating resulted in synchronized sperm beating on some occasions (sperm self-organising behavior). We also found such self-organized sperm behavior on a large scale at the CT in which most of the clustered spermatozoa at the entrance of the intramural UTJ (CT) exhibited synchronized beating (*Figure 4B*; *Figure 4—video 2*). The synchronized sperm beating was observed to generate fluid flows strong enough to prevent other spermatozoa from attaching to the CT or directly pushing out other spermatozoa, thereby preventing other spermatozoa from entering the UTJ (*Figure 4—video 1*; S7).

## Sperm linkage and their migration kinetics

Unlike the sperm trains found in wood mice (*Apodemus sylvaticus*; *Moore et al., 2002*) and deer mice (*Peromyscus maniculatus*; *Fisher and Hoekstra, 2010*), most of the spermatozoa in this study did not form functional sperm trains (linked spermatozoa) (*Figure 5A*; *Figure 5—video 1*). Active sperm linkage or clusters were indeed observed but were rare (only three identifiable cases during over 100 hr of imaging). For the house mouse, the observed sperm trains did not move faster or slower than unlinked single spermatozoa (*Figure 5B*; *Figure 5—video 1*). Their VCL, VSL, and LIN were not faster nor higher than those of unlinked single spermatozoa as shown in *Figure 5B*. Although their SWR may be higher than that of unlinked single spermatozoa, due to the rare number of observed events, further experiments will be necessary to clarify whether the sperm train formation is advantageous in the house mouse.

## Sperm migration and accumulation in the oviduct

After entering the UTJ, spermatozoa continued migration through the narrow UTJ lumen. In the UTJ, spermatozoa interact with epithelia with their hook and penetrate their thin head into a narrow space (*Figure 6—video 1A*). Contraction of the UTJ reduced the width of the UTJ lumen and prevented sperm migration (*Figure 6—video 1B*). However, some spermatozoa could pass through the narrow

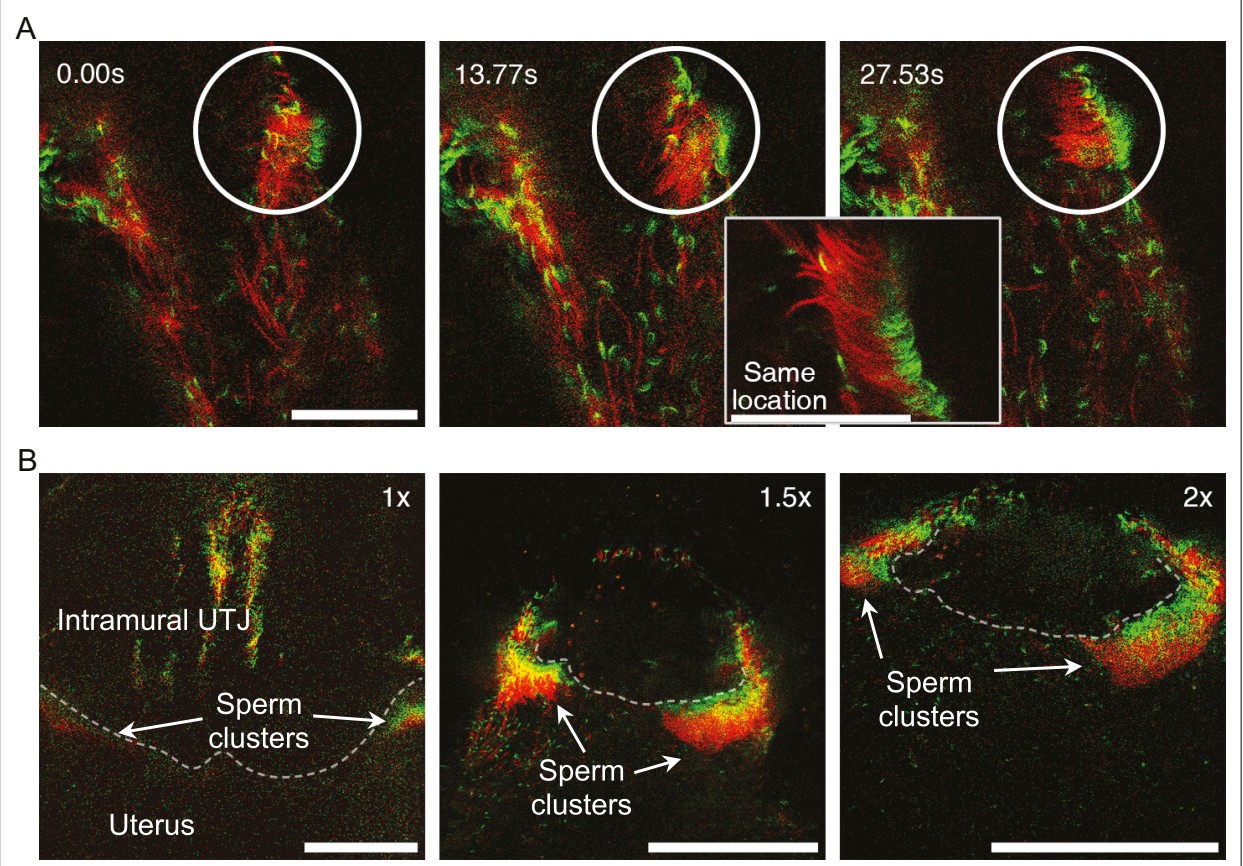

**Figure 4.** Self-organized sperm behavior at colliculus tubarius (CT) inside the uterus. (**A**) The apical shape of the sperm head, due to the sperm hook, results in head asymmetry. This asymmetrical falciform head shape may facilitate sperm re-arrangement and clustering at a uterine wall (***Figure 4—video 1***). The circle highlights spermatozoa undergoing unidirectional re-arrangement over time. The elapsed time after the first frame is shown in the upper left of the images. The right lower zoom-in inset shows an instant of synchronized motion and unidirectional re-arrangement. Scale bar: 50 μm. (**B**) Unidirectional sperm clustering at the entrance of the intramural utero-tubal junction (UTJ) (CT, indicated by a dashed line) is marked with arrows. Such large sperm clustering resulted in synchronized sperm beating at the CT (***Figure 4—video 2***). Scale bar: 200 μm. Note that due to a lack of fluorescence, the principal and terminal pieces of sperm tails are not seen in the images.

The online version of this article includes the following video(s) for figure 4:

**Figure 4—video 1.** Sperm unidirectional re-arrangement in a sperm cluster at a uterine crypt.
https://elifesciences.org/articles/96582/figures#fig4video1

**Figure 4—video 2.** An enormous unidirectional sperm cluster at colliculus tubarius (CT) in the uterus exhibits synchronized sperm beating.
https://elifesciences.org/articles/96582/figures#fig4video2

luminal space by putting their thin hook (***Figure 6A and B***; ***Figure 6—video 1B***). During UTJ lumen contraction, both swimming and beating of spermatozoa therein were physically suppressed. When the UTJ lumen dilates, the suppressed spermatozoa started beating and swimming again (***Figure 6C***; ***Figure 6—video 1B***). We also found sperm accumulations in the oviduct, including UTJ and isthmus (***Figure 6D***). However, as these sperm accumulations were arranged irregularly and consisted of spermatozoa that were mostly acrosome reacted and inactive, they were considered inactive entangled spermatozoa rather than actively linked spermatozoa (or sperm trains). These entangled spermatozoa filled the narrow oviductal lumen, creating an obstruction for the migration of other spermatozoa (***Figure 6—video 1C***). If sperm hook facilitates such entanglement of inactive spermatozoa in the UTJ and isthmus, it can play a role in sperm competition by obstructing migration of the other spermatozoa including those from a second male.

Sperm beating rates of the epithelia-attached spermatozoa were observed to change over time according to the changes in widths of the UTJ lumen (***Figure 6—video 2***). Although such a change in beating rates may simply reflect luminal flow speed or luminal width related to physical space for

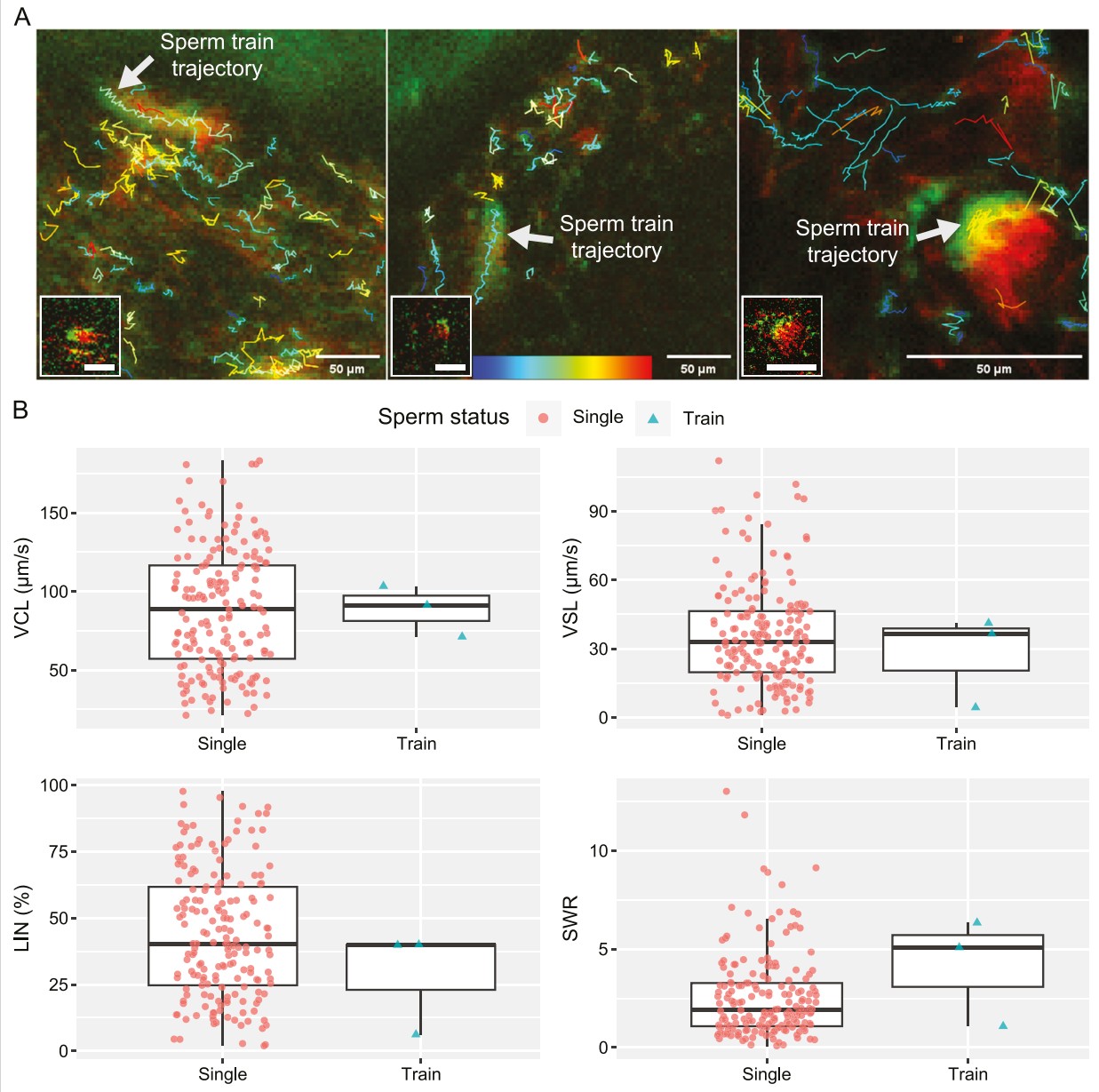

**Figure 5.** Comparative trajectories and kinetics of 3 linked spermatozoa (sperm trains) and 181 unlinked single spermatozoa. (**A**) The projected images, comprising 60 frames, depict the trajectories of sperm trains and unlinked single spermatozoa. Each of the images in the lower left corner shows a sperm train that was traced. *Figure 5—video 1* shows how the three traced sperm trains are moving. The colour bar located at the bottom centre represents the VCL of each sperm trajectory, with blue indicating slower speeds and red indicating faster speeds. (**B**) The boxplots, which include individual data points, represent the kinetic parameters of the sperm trains and unlinked single spermatozoa. The parameters, including curvilinear velocity (VCL), straight-line velocity (VSL), linearity of forward progression (LIN), and straight line-to-sideward movement ratio (SWR), were computed using images of 100x100 pixels that contained a sperm train. The three sperm trains that were traced did not exhibit a faster VCL or VSL, nor a higher LIN. However, it is still possible that the SWR is higher in the sperm train. The lines within the boxes represent the medians, the whiskers represent 1.5 times the interquartile ranges, and the symbols show the individual data points.

The online version of this article includes the following video for figure 5:

**Figure 5—video 1.** Accumulated sperm (sperm trains) are not found to swim faster than unlinked individual sperm in the uterus.

https://elifesciences.org/articles/96582/figures#fig5video1

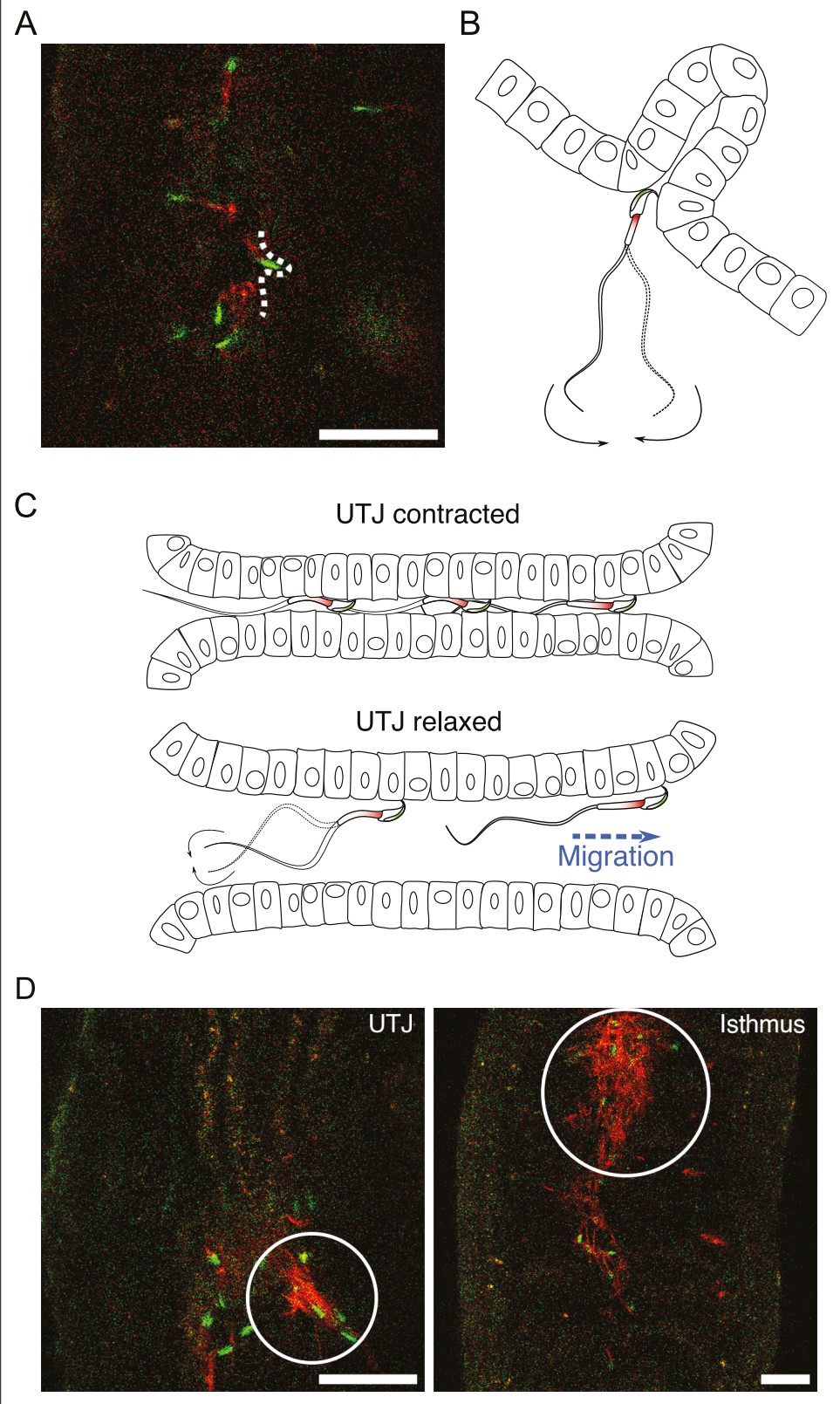

**Figure 6.** Sperm migration through narrow luminal space in utero-tubal junction (UTJ) and various sizes of accumulated sperm in the oviduct. (**A, B**) Spermatozoa can penetrate narrow spaces with their thin head and anchor their hook when they pass through the narrow gaps between mucosal folds during migration in the UTJ (*Figure 6—video 1A*). Scale bar: 50 μm. (**C**) Spermatozoa can only migrate through the UTJ when the luminal

*Figure 6 continued on next page*

*Figure 6 continued*

space is extended due to oviductal contraction and relaxation in the UTJ. (**D**) Entangled spermatozoa in the oviduct, including the UTJ and isthmus, were predominantly made up of inactive acrosome-reacted spermatozoa. These entangled spermatozoa obstruct the migration of other active spermatozoa and can cause damage to live spermatozoa. Scale bar: 50 μm.

The online version of this article includes the following video(s) for figure 6:

**Figure 6—video 1.** Various sperm behaviors in utero-tubal junction (UTJ).

https://elifesciences.org/articles/96582/figures#fig6video1

**Figure 6—video 2.** The beating rate of the attached (anchored) sperm in utero-tubal junction (UTJ) changes over time.

https://elifesciences.org/articles/96582/figures#fig6video2

beating, it may also assist in attaching spermatozoa to the epithelium by providing propulsive forces (*Kantsler et al., 2014*). Fluid flow in the UTJ and isthmus can damage unattached spermatozoa, particularly when they get entangled with inactive entangled spermatozoa that are moving back and forth by the flow (*Figure 6—video 1C*). If such beating prevents epithelium-attached spermatozoa from being swept away by the flow, sperm beating corresponding to the fluid flow will be advantageous. The oviductal fluid gradually flows upward while continuously repeating back-and-forth directional changes and such flow is assumed to aid sperm migration from UTJ to ampulla (*Hino and Yanagimachi, 2019*). However, given our observation of the resistant (anchoring) behavior of spermatozoa to the flow in the UTJ, passive sperm transfer may not always be beneficial for healthy spermatozoa. Additionally, such passive transfer can cause physical damage to the spermatozoa by the collision with other entangled spermatozoa or epithelia of the narrow lumen. Further experiments will be necessary to clarify the role of the rapid upward flow in the oviduct in the fertilization process in mice.

## Discussion

Our real-time deep tissue imaging enabled by two-photon microscopy shows that the mouse sperm hook (i) facilitates sperm interaction with the epithelium for better navigation and (ii) provides an anchor-like role that assists the attachment of sperm to the epithelium. These results suggest that the sperm hook in house mice functions to facilitate sperm migration by interacting with the female reproductive tract (*Firman and Simmons, 2009*; *Suarez, 1987*; *Tourmente et al., 2016*). We also showed that when spermatozoa swim along the uterine wall, their apical hook interacts with the epithelia and determines sperm travelling direction upon encountering uterine epithelium. This finding implies that the sperm hook functions as a pivot when spermatozoa reach the uterine epithelium, and help spermatozoa migrate along the uterine wall by assisting in pro-wall-hook sperm orientation. Additionally, the sperm hook plays an important role in resisting endogenous fluid flow in the female reproductive tract by providing an anchor-like function. We did not observe a linkage of spermatozoa (sperm trains) that enables faster and straighter swimming of the sperm train in the uterus or oviduct. Instead, we found instances of the entanglement of inactive spermatozoa in the oviduct. Such aggregates of inactive spermatozoa appear to obstruct the migration of other spermatozoa. However, the role that these entangled spermatozoa play in sperm competition within the oviduct is a subject that necessitates further investigation.

Previous studies using various murine rodents suggest that acquiring the ability to form sperm train is a relatively rare evolutionary event in this taxon (*Tourmente et al., 2016*; *Varea-Sánchez et al., 2016*). Instead of finding evidence that the sperm hook aids in sperm train formation, those studies showed that variations in the length of the sperm hook, and asymmetry of sperm head influenced by the sperm hook are two important variables that reflect the degree of the sperm competition in murine rodents. One study even suggested that the sperm train formation (sperm linkage) is independent of the existence of sperm hook (*Tourmente et al., 2016*). Therefore, it is questionable whether acquiring the sperm hook in the murine rodents has evolved in general for a better sperm linkage. In line with the previous studies, our current study also showed that sperm hook plays a pivotal role in sperm-epithelium interactions by aiding sperm attachment to the uterine and oviductal epithelium and influencing sperm orientation during migration inside the female reproductive tract (*Figure 1E and F*; *Figure 1—video 3A, B*). Such a role of the sperm hook suggests that the hook facilitates

interactions between sperm and epithelia of the female reproductive tract and supports the migration hypothesis.

As we did not observe any passive sperm transfer by the internal fluid flow from the uterus to intramural UTJ through the CT, it is puzzling how spermatozoa enter the intramural UTJ. Although we proposed a hypothetical model of sperm passage through the CT in which the sperm hook plays an important role by providing an anchor-like role that prevents backward movement but allows head-forward sliding, further investigation with new experimental and observational tools will be necessary. Nevertheless, our findings suggest that the CT is a key female anatomical structure that controls sperm entry to the UTJ and influences sperm competition. If sperm passage through the CT is too easy, associated risks such as polyspermy or pathogen transmission that incur fitness costs for females also increase (*Firman and Simmons, 2013*; *Mahabir et al., 2008*). Therefore, the number of sperm passing through the CT should be balanced by the conflict between the two sexes depending on species-specific mating systems and the degree of promiscuity. This conflict then gives rise to an evolutionary arms race, with the male's sperm fertilization ability on one side, which includes aspects such as sperm swimming speed, and the female's sperm selection process on the other, which encompasses elements like physio-chemical barriers in the female reproductive tract (*Firman et al., 2017*; *Lüpold and Pitnick, 2018*; *Simmons and Wedell, 2020*). Therefore, species variations in the length of the sperm hook and head asymmetry in rodents suggested in the previous studies (*Tourmente et al., 2016*; *Varea-Sánchez et al., 2016*) will also reflect the mechanical and structural characteristics of the CT and lumen in the intramural UTJ.

In the current study, we could not confirm evidence of sperm cooperation via sperm train formation that aids faster sperm swimming. Instead, we discovered a new form of potential sperm cooperation – synchronized sperm beating that resulted from spontaneous unidirectional sperm clustering. Based on these observations, we propose that the asymmetry of the mouse sperm head with an apical hook plays a crucial role in synchronized sperm beating (sperm cooperation) by facilitating unidirectional sperm re-arrangement at the uterine wall. The asymmetrical head shape in the house mouse, therefore, may have evolved not only to facilitate sperm migration but also to facilitate such sperm self-organized behaviors including unidirectional clustering and following beating synchrony. Under this scenario, sperm cooperation in house mice is not only mediated by sperm train formation as in some rodent species (*Fisher and Hoekstra, 2010*; *Moore et al., 2002*), but mediated by the synchronized sperm beating that obstructs migration of other spermatozoa in the uterus. Unidirectional sperm clustering and its potential role in sperm migration were also suggested in a previous study using fixed and tissue-cleared samples with in-vitro live sperm analysis (*Qu et al., 2021*). We could not find any supporting evidence that such clustering helps sperm passage through the CT. However, our observations regarding self-organized sperm behavior provide insights into the evolution of the asymmetrical structure of the sperm head in rodent species. These findings present an opportunity to reconcile two hypotheses: the cooperation hypothesis (*Fisher and Hoekstra, 2010*; *Immler et al., 2007*; *Moore et al., 2002*) and the migration hypothesis (*Firman and Simmons, 2009*; *Smith and Yanagimachi, 1990*).

The evolution of sperm characteristics, including the sperm hook, is a complex process influenced by several factors. For instance, sperm competition between ejaculates is a significant driving force influencing sperm head morphology and behavior in rodents (*Fisher et al., 2014*; *Immler et al., 2007*; *Tourmente et al., 2016*; *Varea-Sánchez et al., 2016*). Cryptic female choice also plays a crucial role in the evolution of sperm head shape and sperm kinetic characteristics (*Birkhead, 1998*; *Eberhard and Lehmann, 2019*; *Firman et al., 2017*). Further investigation of sperm behavior inside the female reproductive tract or tissue mimicking microfluidic devices with real-time deep tissue imaging as in the current study, will provide valuable opportunities for a more comprehensive examination of both sperm-sperm and sperm-epithelium interactions in the female reproductive tract. While we have focused on observing sperm interactions for only naturally healthy mice in this study, future works employing specifically targeted genetically modified knockout animal models will further elucidate and confirm the exact genetic and functional mechanisms that guide these interactions. This will help us better understand not only sperm competition and cryptic female choice in mice, but also those in other animals, including humans. While current assessments of sperm health generally involve measuring sperm count, movement, and shape, the current study suggests that analysing interactions between spermatozoa and the female reproductive tract is important and warrants further exploration.

Given the significance of sperm health for fertility, this work not only highlights the importance of interactions between male sperm and female reproductive tract in successful migration, but also opens new avenues for understanding different causes of infertility and possible targets for treatment.

## Materials and methods

### Custom-built two-photon microscope

To observe sperm behavior in the female mouse reproductive organ, we built a video rate (30 frames/second at 512 x 512-pixel resolution) Two-Photon Laser Scanning Fluorescence Microscope (2PLSM; *Figure 1—figure supplement 1*). A tunable femtosecond pulse laser (Chameleon, Discovery) was tuned to a choice of wavelengths from 960, 970, 980, and 1000 nm to simultaneously excite GFP and DsRed for sperm imaging and for autofluorescence imaging of the reproductive tract. Imaging quality was found to be similar for these wavelength ranges. All images were taken using a water dipping low magnification high NA objective lens (Nikon 16 X, 0.8NA). Video-rate imaging was achieved using a resonant Galvo scanning mirror system oscillating at 8 kHz. The laser power was actively controlled using a Pockels cell. Synchronization between the Galvos, sample/objective stages, Pockels cell, photomultiplier tubes (PMTs), and the data acquisition systems were controlled using ScanImage, ver. SI2021.1.0 (*Pologruto et al., 2003*).

### Mice preparation and mating

We used two male transgenic mice lines for mating experiments. We purchased B6D2-Tg(CAG/Su9-DsRed2, Acr3-EGFP)RBGS002Osb male mice that express DsRed in the mitochondria at sperm midpiece and EGFP in the sperm acrosome from Riken BRC, Japan, depositor: M. Ikawa (*Hasuwa et al., 2010*). We then conducted in vitro fertilization to produce specific pathogen-free (SPF) F1 mice. The fertilized eggs (2 cell stage embryos) were then artificially inseminated into SPF wild-type C57BL/6 J females. After we confirmed the successful production of transgenic F1 male mice by PCR, we confirmed the SPF status and formed two breeding colonies with the transgenic F1 males under the SPF condition. One breeding colony comprised two wild-type C57BL/6 J females to better reproduce F2 generation. We also made breeding colonies that consisted of a transgenic C57BL/6 J female, Cx3cr1tm2.1(cre/ERT2)Litt/WganJ (JAX stock #021160, Cx3cr1 female) that expresses EYFP in microglia (*Parkhurst et al., 2013*) to test whether sperm functionality changes in other mice including double-transgenic mice. When F2 mice got older than 6 weeks, they were transferred to another room where mating experiments were conducted. After transfer, each male mouse for mating experiments was single-caged. We used the F2 males that derived from both colonies that had the two genes (CAG/Su9-DsRed2 and Acr3-EGFP; RBGS male) or three genes (CAG/Su9-DsRed2, Acr3-EGFP, and Cx3cr1; RBGS-Cx3cr1 male) for mating experiments. We could not find any phenotypic difference in the sperm of the two strains – sperm from both strains expressed red fluorescence at the midpiece and green fluorescence at the acrosome at the head. We confirmed that both strains of F2 males were fertile, and their sperm also successfully migrated through the female reproductive tract, from the uterus to the ampulla.

In total, we used 3 males that successfully mated with females due to space limits in the experimental room (*Supplementary file 1a*). The 3 males were used repeatedly for all mating experiments in the current study except one vasectomized RBGS-Cx3cr1 male that was used only once for comparison of the CT structure for virgin and non-virgin female mice (*Supplementary file 1b*). All mice were kept under a housing condition that allows free access to food and water with 12 hr of light and dark cycle (lighting from 6 to 18 o'clock, dark from 18 to 6 o'clock). Mating experiments were done under light conditions from 9 am to 12 pm for 3 hr. Oestrus was induced by exposing male bedding materials (wood shavings) that consisted of male excretion to females older than 8 weeks. Three to seven days after exposure to the bedding materials, oestrus was checked daily following previously established protocols (*Byers et al., 2012*). When we found oestrous females, we relocated one or two females to a single-caged male. When males showed no interest in the female (no mounting attempts) or the female rejected the male's mounting attempt for the first 10 min, we returned the female to its original cage. The returned females were not exposed to other males until the next mating trial on the next day or one week later. All females in the experiments had no birth records before successful copulatory plug-confirmed mating. However, some of them probably had multiple oestrous cycles given our

multiple oestrus-inducing trials. We did not limit the age of females and males for our experiments to minimize the number of sacrificed animals. We observed the male's mating until we could observe ejaculation. To confirm male ejaculation, we checked the copulatory plug from the female genitalia after we observed ejaculatory behavior – the male stops thrusting and holds the female for about 5–10 s when it ejaculates. After this ejaculatory behavior, we waited for 2 min and if the male did not exhibit further mounting, we checked the copulatory plug from the female genitalia. When the male ejaculated, we kept them together for up to 3 hr then took out the female for imaging experiments.

## Ex vivo imaging with two-photon microscopy

Female mice were sacrificed by cervical dislocation after anaesthesia using 2% isoflurane inhalation which usually took less than 5 min. After euthanasia, the female reproductive tract with the copulatory plug was excised and washed with Dulbecco's modified Eagle's medium (*Tayama et al., 2006*) (DMEM; GibcoTM, cat. No. 21063029). After washing, the reproductive tract was attached to a tissue culture dish with tissue adhesive (3 M Vetbond 1469 SB). After attachment, we filled the dish with 37°C preheated medium that contained an equal amount of DMEM and modified human tubal fluid (mHTF; Fujifilm Irvine Scientific, cat. ID. 90126) medium. All media were stored for at least 1 hr in a 37°C preheated incubator with 5% of $CO_2$ concentration before use. The culture dish was then placed on the 37°C preheated metal mount of the two-photon microscope and imaged with varying laser power for different depths. Most of the images were taken with 512×512 pixel image size at 30 fps (262,144 pixels per frame). Multicolour imaging was performed where each frame of the image has two colour channels (red and green). If needed, we could increase the frame rate up to 110 fps or higher by reducing the acquired image size to 128x128 (16,384 pixels per frame). We conducted observation for about 3–6 hr. During our observation, the uterus continued contraction and relaxation cycles.

To image the entire depth of the reproductive tract, we applied tissue clearing to investigate the structure of the UTJ entrance (or colliculus tubarius, CT) using the C-Match solution (RI = 1.46, Crayon technologies, Korea). In brief, the tissue was first washed three times in PBS and fixed in 4% paraformaldehyde for 3 hr at 4°C in a refrigerator. After fixation, we washed the tissue three times with PBS and the absorbed residual PBS using paper towels. We then added C-Match to the sample waited overnight and imaged it on the next day with the sample submerged in C-Match. We imaged three cleared samples from three different females. In one sample (*Figure 3A*), only the intramural UTJ part was excised from one unmated transgenic C57BL/6 J female mouse – a hybrid female that was delivered from Cx3cr1 female and Thy1 male (JAX stock #030526) called Tg(Thy1-jRGECO1a)GP8.31Dkim (*Dana et al., 2018*). Another sample was from a female that was mated with a vasectomized RBGS-Cx3cr1 male. In this sample, the whole female reproductive tract was excised with the copulatory plug, so the intramural UTJ was covered by the uterus. This sample was used to compare the UTJ for virgin and non-virgin females. The final sample was from a wild-type female that was mated with an RBGS-Cx3cr1 male. For cleared tissue imaging, we acquired 3D volume images with 2 um Z-axis step size while averaging 20 (first sample) or 30 images (second sample) per slice to increase the signal-to-noise ratio using autofluorescence of the reproductive tract tissue.

## Sperm tracking and speed measurement

We used Fiji (*Schindelin et al., 2012*) to process acquired images and its plugin, called TrackMate (*Ershov et al., 2022*; *Tinevez et al., 2017*) to track sperm trajectory in the uterus. We extracted trajectories from 60 sequential images (duration 2 s) for sperm tracking when uterus movement was the smallest. We also used Turboreg (*Thévenaz et al., 1998*), an ImageJ plugin, to realign the images when there was a shift between images due to uterine movement. Additionally, out of eight stacked images, we cropped three to obtain a straight view of the uterine wall. After preparing the images, we targeted the sperm head to track sperm as the sperm head expressed EGFP which was easy to track. We used Thresholding Detector to select sperm heads and LAP Tracker to trace sperm trajectories using the TrackMate plugin. We also adjusted parameter values in the plugin to better select sperm trajectories. Our final parameter values are as follows: head radius (>0.75 μm), frame-to-frame linking (10~11 μm), track segment gap closing (max distance: 10~11 μm, max frame gap: 2), number of spots in track (>6), and max distance travel (2.5 μm). When there were artificial trajectories that were not from sperm, we manually removed the track. If the original parameters could not detect well or had too many false tracks, we adjusted two parameters; frame-to-frame linking (up to 12 μm), and track

segment gap closing (only max distance up to 12 μm). We also tracked the trajectories of sperm trains using the same parameters and settings. However, to reduce computation time and prevent mis-tracking of non-sperm cells, we cropped the images and utilized 100×100 pixel images that contain the entire trajectory of each sperm train as well as other unlinked single spermatozoa.

To calculate sperm speed in relation to the distance from the uterine wall, we need to define the uterine wall. To define the uterine wall, we first selected images with a straighter uterine wall and projected the extracted 60 sequential images into one plane by taking the maximum intensity projection with some adjustment of brightness and contrast using Fiji. For some images taken at a low magnification level that contained curved uterine walls, we used parts of the field of view that contained straight walls appropriate for our analysis. Next, we used the object selection tool of Adobe Photoshop CC (23.1.0 version) to automatically select the walls from the projected images. We then extracted the uterine wall image layer and pasted it to a blank image with white background. Finally, we extracted the wall coordinates by selecting the non-zero-valued pixels that formed the boundary of the uterine wall (blue-coloured area in *Figure 2—figure supplement 1*). The boundary coordinates were converted to a micrometre scale based on the magnification and image resolution to normalize the units between different images obtained with different magnification factors. We fitted the uterine wall coordinates using linear regression. The fitted linear regression line was considered the uterine wall of each female ((1) in *Figure 2—figure supplement 1*) and used to calculate the distance and angle (radian) between sperm and the wall.

We measured the distance between sperm and the uterine wall by calculating the minimum distance between the mid-point of each sperm trajectory and the fitted line of the uterine wall ((2) in *Figure 2—figure supplement 1*). The angle between a sperm trajectory and the uterine wall was calculated by measuring the angle between the fitted line and a straight line that passed the first and last spots of the trajectory ((3) in *Figure 2—figure supplement 1*). We then computed sperm progression speed parameters used in CASA (*Amann and Waberski, 2014*) using our sperm tracking data using the TrackMate plug-in. We first calculated curvilinear velocity, VCL by dividing the total distance travelled (μm) by total track time – the time (second) taken from the first point (spot) to the last point in a sperm trajectory. The straight line velocity, VSL was calculated by dividing the track displacement – the distance between the first and last spots of a sperm trajectory – by the total track time (second). We also calculated the linearity of forward progression, and LIN of sperm by dividing VSL by VCL (range 0–1). Along with the above CASA-used parameters, we defined a new parameter, the SWR to estimate sperm migration linearity by comparing forward and sideward moving distances. SWR was calculated by dividing the track displacement (μm) of a sperm trajectory by the maximum sideward movement distance (μm) – the maximum distance between two parallel lines passing each point (spot) that parallel to the track displacement line ((4) in *Figure 2—figure supplement 1*). Further information on the terms and parameters of the TrackMate plug-in used in the current paper is also described in a paper and manual by the developers (*Ershov et al., 2022*; *Tinevez et al., 2017*).

## Statistical analysis

All statistical analyses were done using R, version 4.3.0 (*R Development Core Team, 2023*). To estimate sperm swimming speed and linearity in relation to the uterine wall, we used data from eight copulation experiments between eight females and three males. We ran four generalized linear mixed models to test whether sperm move faster and straighter when they migrate along the uterine wall. We used log-transformed VCL, VSL, LIN, and SWR as response variables in each model. In all models, we included the angle and distance between the wall and sperm trajectories as explanatory variables. We also included whether we cropped the image or not (O or X) to check the effect of image cropping on the analysis. Male IDs and the date of experiments were included as random effects in all models to control possible individual variations in sperm and reproductive tract properties. All models did not violate assumptions. All full models were also compared with null models that only included random effect and all full models were significantly better than the null models. All variance influencing factors (VIFs) were less than 1.1 which indicates no serious drawback from collinearity.

## Acknowledgements

We appreciate members of the Bio-Optics Lab and In-vivo Research Center at the Ulsan National Institute of Science and Technology for their help in building and maintaining microscopy instruments

and mouse strains; particularly I Kim, S Park, Y Lee, Y Kwon, Y Choi, H Kim, E Cho, and T Asadishad. We also thank M Okabe, G Kang, and Y Kawaguchi for their comments about the methods and manuscript. Funding sources: National Research Foundation of Korea grant 2020R1A6A3A01098226 (HR), National Research Foundation of Korea grant 2019M3E5D2A01063812 (JP), National Research Foundation of Korea grant 2021R1A2C3012903 (JP), National Research Foundation of Korea grant 2021R1A4A1031644 (JP), National Research Foundation of Korea grant RS-2023–00264980 (JP), Ministry of Science and ICT (IITP-2023-RS-2023–00259676) (JP), National Research Foundation of Korea grant 2021M3A9G8022960 (JK1, Jae-Ick Kim), National Research Foundation of Korea grant 2022M325E8017907 (JK1)

## Additional information

### Funding

| Funder | Grant reference number | Author |
|---|---|---|
| National Research Foundation of Korea | 2020R1A6A3A01098226 | Heungjin Ryu |
| National Research Foundation of Korea | 2019M3E5D2A01063812 | Jung-Hoon Park |
| National Research Foundation of Korea | 2021R1A2C3012903 | Jung-Hoon Park |
| National Research Foundation of Korea | 2021R1A4A1031644 | Jung-Hoon Park |
| National Research Foundation of Korea | RS-2023-00264980 | Jung-Hoon Park |
| Ministry of Science and ICT, South Korea | IITP-2023-RS-2023-00259676 | Jung-Hoon Park |
| National Research Foundation of Korea | 2021M3A9G8022960 | Jae-Ick Kim |
| National Research Foundation of Korea | 2022M325E8017907 | Jae-Ick Kim |

The funders had no role in study design, data collection and interpretation, or the decision to submit the work for publication.

### Author contributions

Heungjin Ryu, Conceptualization, Data curation, Formal analysis, Funding acquisition, Investigation, Visualization, Methodology, Writing – original draft, Writing – review and editing; Kibum Nam, Investigation, Methodology; Byeong Eun Lee, Investigation; Yundon Jeong, Software, Investigation, Visualization, Methodology; Seunghun Lee, Jeongmo Kim, Young-Min Hyun, Methodology; Jae-Ick Kim, Supervision, Funding acquisition, Project administration, Writing – review and editing; Jung-Hoon Park, Conceptualization, Supervision, Funding acquisition, Investigation, Visualization, Methodology, Writing – original draft, Project administration, Writing – review and editing

### Author ORCIDs

Heungjin Ryu ⓘ https://orcid.org/0000-0003-1048-9519
Young-Min Hyun ⓘ https://orcid.org/0000-0002-0567-2039
Jae-Ick Kim ⓘ https://orcid.org/0000-0001-9705-0394
Jung-Hoon Park ⓘ https://orcid.org/0000-0001-5316-1690

### Ethics

All procedures comply with the guidelines for standard operations set by the Institutional Animal Care and Utilization Committee (IACUC) at the Ulsan National Institute of Science and Technology (UNIST). These procedures have been approved by the IACUC of UNIST under Permit Numbers: UNISTIACUC-20-41 and UNISTIACUC-21-39.

Reviewer #1 (Public review): https://doi.org/10.7554/eLife.96582.4.sa1
Author response https://doi.org/10.7554/eLife.96582.4.sa2

## Additional files

### Supplementary files

• Supplementary file 1. Tables with experiment animal information and results of the generalized linear mixed models (GLMM) analysis. (a) Basic information of 4 males that were used for the mating experiment. (b) Mating records and the information of the females for sperm tracking. (c) Summary results of the GLMM.

• MDAR checklist

### Data availability

Raw data on sperm trajectories in the female reproductive track are available at figshare. Custom codes for sperm tracking data analysis written with Matlab (version R2019b) are available at GitHub (copy archived at *Jeong, 2023*).

The following dataset was generated:

| Author(s) | Year | Dataset title | Dataset URL | Database and Identifier |
|---|---|---|---|---|
| Ryu H, Nam K, Lee BE, Jeong Y, Lee S, Kim J, Hyun YM, Kim JI, Park JH | 2024 | Data from: Mouse sperm hook facilitates sperm anchoring and probing for successful migration | https://doi.org/10.6084/m9.figshare.22878230 | figshare, 10.6084/m9.figshare.22878230 |

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
