## [Editor Report · eLife Assessment]

This study uses ex vivo live imaging of uteri post-mating to test the role of the sperm hook in the house mouse sperm in sperm movement that would be interesting to evolutionary biologists. The significance of the work is **useful** as live imaging can reveal information not seen in fixed images. The strength of evidence is **incomplete** as they cannot directly test the role of the sperm hook in facilitating movement along the uterine wall.

---

## [Referee Report · Reviewer #1 (Public review)]

Summary:

The authors want to determine the role of the sperm hook of the house mouse sperm in movement through the uterus. They use transgenic lines with fluorescent labels to sperm proteins, and they cross these males to C57BL/6 females in pathogen-free conditions. They use 2-photon microscopy on ex vivo uteri within 3 hours of mating and the appearance of a copulation plug. There are a total of 10 post-mating uteri that were imaged with 3 different males. They provide 10 supplementary movies that form that basis for some of the quantitative analysis in the main body figures. Their data suggest that the role of the sperm hook is to facilitate movement along the uterine wall.

Strengths:

Ex vivo live imaging of fluorescently labeled sperm with 2-photon microscopy is a powerful tool for studying the behavior of sperm.

Weaknesses:

The paper is descriptive and the data are correlations.

The authors cannot directly test their proposed function of the sperm hook in sliding and preventing backward slipping.

---

## [Author Response]

The following is the authors’ response to the previous reviews.

**Public Reviews:**

**Reviewer #1 (Public Review):**
Summary:The authors want to determine the role of the sperm hook of the house mouse sperm in movement through the uterus. They use transgenic lines with fluorescent labels to sperm proteins, and they cross these males to C57BL/6 females in pathogen-free conditions. They use 2-photon microscopy on ex vivo uteri within 3 hours of mating and the appearance of a copulation plug. There are a total of 10 post-mating uteri that were imaged with 3 different males. They provide 10 supplementary movies that form the basis for some of the quantitative analysis in the main body figures. Their data suggest that the role of the sperm hook is to facilitate movement along the uterine wall.Strengths:Ex vivo live imaging of fluorescently labeled sperm with 2-photon microscopy is a powerful tool for studying the behavior of sperm.Weaknesses:The paper is descriptive and the data are correlations.The authors cannot directly test their proposed function of the sperm hook in sliding and preventing backward slipping.
**Recommendations for the authors:**

**Reviewer #1 (Recommendations For The Authors):**
I suggest that the authors clearly state and explain in the manuscript that this study is limited with respect to the ability to "directly test the role of the sperm hook in facilitating movement along the uterine wall". I think that if they make this statement in the manuscript, perhaps at the end of the abstract, then the strength of evidence for their claims could be deemed as solid after re-review.

We thank the reviewer again for the review process. We believe that our manuscript has improved considerably during the review process. Regarding the limitations and future work, we have added the following to the discussion section.

“Further investigation of sperm behaviour inside the female reproductive tract or tissue mimicking microfluidic devices with real-time deep tissue imaging as in the current study, will provide valuable opportunities for a more comprehensive examination of both sperm-sperm and sperm-epithelium interactions in the female reproductive tract. While we have focused on observing sperm interactions for only natural healthy mice in this study, future works employing specifically targeted genetically modified knockout animal models will further elucidate and confirm the exact genetic and functional mechanisms that guide these interactions.”

The revised manuscript is an improvement over the initial submission. I suggest that the authors mark the oviduct explicitly in Fig. 1A.

The oviduct includes the ampulla, isthmus, and UTJ. We have additionally marked the oviduct in Fig. 1A, with according arrows and a box.